# Implicit Ensemble Training for Efficient and Robust Multiagent Reinforcement Learning

**Macheng Shen**                                                                 *macshen@mit.edu*
*Department of Mechanical Engineering*
*Massachusetts Institute of Technology*

**Jonathan P. How**                                                              *jhow@mit.edu*
*Department of Department of Aeronautics and Astronautics*
*Massachusetts Institute of Technology*

Reviewed on OpenReview: *https://openreview.net/forum?id=LfTukxzxTj&referrer=%5BTMLR%5D(%2Fgroup%3Fid%3DTMLR)*

## Abstract

An important issue in competitive multiagent scenarios is the distribution mismatch between training and testing caused by variations in other agents' policies. As a result, policies optimized during training are typically sub-optimal (possibly very poor) in testing. Ensemble training is an effective approach for learning robust policies that avoid significant performance degradation when competing against previously unseen opponents. A large ensemble can improve diversity during the training, which leads to more robust learning. However, the computation and memory requirements increase linearly with respect to the ensemble size, which is not scalable as the ensemble size required for learning robust policy can be quite large (Czarnecki et al., 2020). This paper proposes a novel parameterization of a policy ensemble based on a deep latent variable model with a multi-task network architecture, which represents an ensemble of policies implicitly within a single network. Our implicit ensemble training (IET) approach strikes a better trade-off between ensemble diversity and scalability compared to standard ensemble training. We demonstrate in several competitive multiagent scenarios in the board game and robotic domains that our new approach improves robustness against unseen adversarial opponents while achieving higher sample-efficiency and less computation.

## 1  Introduction

In competitive multiagent scenarios, agents learn concurrently (Lowe et al., 2017) and the learned policy of each agent depends on the joint policy of all the other learning agents. One challenge resulting from such multiagent learning is that the policy learned from training might not perform well in the testing environment where the opponents' policies could be significantly different from those of the training opponents (distribution shift between training and testing). Even worse, the testing opponents could be trained to exploit the weakness of the policy learned from the training (Gleave et al., 2019).

One effective approach to mitigate the performance degradation from training to testing is ensemble training, which has been applied in many previous works (Bansal et al., 2017; Lowe et al., 2017; Silver et al., 2016; Jaderberg et al., 2019; Vinyals et al., 2019) and shown improved robustness against previously unseen opponents compared with training a single policy per agent. In ensemble training, each agent has multiple policies as in (Lowe et al., 2017; Jaderberg et al., 2019; Vinyals et al., 2019) or keeps multiple copies of previous policies as in (Silver et al., 2016; Bansal et al., 2017), from which one policy for each agent is sampled to play against each other. As a result, each policy is optimized against a distribution of the other agents' policies, which effectively reduces the distribution shift between training and testing (because

a single policy can be regarded as a single Dirac-delta distribution centered at one point within the policy space). However, one drawback of applying this ensemble training technique is the significant increase in computation and memory consumption due to the learning and storage of multiple policies. Besides, the number of policies required for guaranteed policy strength improvement is a function of the non-transitivity dimension of the multiagent scenario, which could be on the order of tens for simple games such as Tic-Tac-Toe and more than thousands for complicated real-world games (Czarnecki et al., 2020). As a result, this ensemble training approach could become intractable due to constraints on computational resources and/or end up learning weak policies due to insufficient policy diversity.

In this paper, we propose a novel implicit ensemble training (IET) approach by formulating ensemble training as a multitask learning problem. [1] Instead of maintaining multiple policies explicitly with independent neural networks, our IET approach uses a multitask network architecture with a learnable conditional latent variable distribution to implicitly represent a policy ensemble. The multitask network architecture improves sample-efficiency via parameter sharing and the conditional latent variable captures the diversity of the policy.

Our contributions are: 1) we identify the cause of inefficiency in standard ensemble training; 2) we develop a novel algorithm that extends ensemble training with latent variables and a multitask network, resulting in a better trade-off between policy diversity and scalability, which is demonstrated by improved policy robustness with fewer samples and less computation compared with standard ensemble training.

## 2 Preliminaries

### 2.1 Markov games

We use Markov Games (Littman, 1994) as the decision-making framework in our work. A Markov game for $N$ agents is defined by a set of states $\mathcal{S}$ describing the possible configurations of all agents, a set of actions $\mathcal{A}_1, \ldots, \mathcal{A}_N$, and a set of observations $\mathcal{O}_1, \ldots, \mathcal{O}_N$ for each agent. Each agent has a stochastic policy $\pi_i : \mathcal{O}_i \times \mathcal{A}_i \mapsto [0, 1]$, and a reward function $r_i : \mathcal{S} \times \mathcal{A}_i \mapsto \mathbb{R}$.

### 2.2 Multiagent reinforcement learning (MARL)

The objective of each agent is to maximize its own cumulative reward $R_i = \sum_{t=0}^{T} \gamma^t r_i^t$ with discount factor $\gamma$ and time horizon $T$ (Lowe et al., 2017). As a result, the learning problem is formulated as finding a joint policy $\boldsymbol{\pi} = \{\pi_i\}^{i=1:N}$, where each policy maximizes its expected cumulative reward,

$$J_i = \mathbb{E}_{s \sim p^{\boldsymbol{\pi}}, a_i \sim \pi_i, \boldsymbol{a}_{-i} \sim \boldsymbol{\pi}_{-i}} \left[ R_i(s, \boldsymbol{a}) \right], \tag{1}$$

with $p^{\boldsymbol{\pi}}$ being the transition dynamics and the subscript $-i$ denotes the set $\{j | j \neq i, j = 1, 2, \ldots, N\}$.

### 2.3 MARL with policy distribution

In practice, learning with a single joint policy typically leads to over-fitting between policies and poor generalization to previously unseen policies. A variety of empirical studies (Lowe et al., 2017; Silver et al., 2016; Jaderberg et al., 2019; Vinyals et al., 2019; Lanctot et al., 2017; Bansal et al., 2017; Vinitsky et al., 2020) and a recent theoretical study (Czarnecki et al., 2020) suggest that it is necessary to maintain a diverse set of policies for each agent to improve the strength of the joint policy learned via MARL. Therefore, instead of learning a single joint policy, we consider the following objective function that learns a distribution of policies for each agent,

$$J_i = \mathbb{E}_{s \sim p^{\boldsymbol{\pi}}, \boldsymbol{a} \sim \boldsymbol{\pi}, \boldsymbol{\pi} \sim \mathcal{P}(\boldsymbol{\Pi}),} \left[ R_i(s, \boldsymbol{a}) \right], \tag{2}$$

where $\mathcal{P}$ is a joint distribution over the joint policy space $\boldsymbol{\Pi} = \Pi_1 \times \Pi_2 \ldots \times \Pi_N$. Each agent is learning its own policy distribution $\Pi_i$ to optimize its objective $J_i$ subject to the joint distribution $\boldsymbol{\Pi}$.

---

[1]code: https://github.com/MachengShen/ImplicitEnsembleTraining

Note that the feasibility set of Eq. 2 contains that of Eq. 1, which is analogous to the relationship between a mixed-strategy Nash Equilibrium and a pure-strategy Nash Equilibrium (Nash et al., 1950). This relationship also suggests that Eq. 2 is a more general learning objective than Eq. 1.

One simple parametrization of $\mathcal{P}_j(\Pi_j)$ is a uniform distribution over a policy ensemble of fixed size (a set of independent policies, each parameterized by a neural network), such as in (Lowe et al., 2017; Bansal et al., 2017; Vinitsky et al., 2020). Policy space response oracle (PSRO) (Lanctot et al., 2017) is another parametrization, with an expanding set of policies. At each iteration, a new policy produced by an oracle, via solving for the (approximate) best response against a mixture of other agents' policies, is added to the policy ensemble. PSRO leads to better convergence towards Nash Equilibrium in poker games than self-play (single policy) and fictitious self-play (uniform distributed over previous policies). Despite the better convergence property, these ensemble approaches are not scalable because the computation increases linearly with respect to the ensemble size, which can be exponential with respect to the complexity of the multiagent interactions (Czarnecki et al., 2020) to guarantee policy improvement. This limitation motivates developing a new parametrization of policy distribution with better parameter-efficiency, which leads to higher sample-efficiency for training.

## 3 New approach

In this section, we present a parameter-efficient parametrization of the policy distribution, which we call an implicit ensemble. We start by discussing the relationship between a uniform ensemble with a latent-conditioned policy and then show how our implicit ensemble approach extends the uniform ensemble for higher parameter-efficiency and policy diversity.

### 3.1 Generalization of ensemble training as latent-conditioned policy

Ensemble training with a uniform distribution over a fixed-sized set of policies is a common approach, e.g. in (Lowe et al., 2017; Bansal et al., 2017; Vinitsky et al., 2020), for improving the robustness of policies in MARL. In ensemble training, each agent's policy $\pi_i$ is an ensemble of $K$ sub-policies, with each denoted as $\pi_{\theta_i^{(k)}}$ or $\pi_i^{(k)}$ and parameterized by separate neural network parameters $\theta_i^{(k)}$. The generative process for the ensemble training policy $\pi_i$ is expressed as

$$
\begin{aligned}
k &\sim \text{unif}(1, \text{K}) = \frac{1}{K} \sum_{j=1}^{K} \delta(k - j), \\
\theta_i | k &\sim \delta(\theta_i - \theta_i^{(k)}), \\
a_i | \theta_i &\sim f_{\theta_i}(o_i),
\end{aligned}
\tag{3}
$$

where $\delta$ is the Dirac-delta distribution. One interpretation of Eq. 3 is that $\pi_i$ is a conditional policy on a uniform discrete latent random variable $k$,

$$
a_i | k \sim \bar{f}(o_i, \theta_i^{(1)}, \theta_i^{(2)}, \ldots, \theta_i^{(K)}; k) = f_{\theta_i^{(k)}}(o_i).
\tag{4}
$$

Eq. 4 suggests that ensemble training can be interpreted as a latent-conditioned policy (Haarnoja et al., 2018; Petangoda et al., 2019; Lee et al., 2019) conditioned on the discrete random variable $k$ coming from a uniform distribution. However, this parametrization is inefficient because only one out of the $K$ sets of sub-policy parameters $[\theta_i^{(1)}, \theta_i^{(2)}, \ldots, \theta_i^{(K)}]$ is activated per sampling of $a_i$. As a result, the rollout from executing one set of sub-policy parameter cannot be used to optimize the rest $K - 1$ sets of sub-policy parameters, which reduces the sample efficiency by a factor of $K$.

### 3.2 Implicit ensemble training

Our IET generalizes ensemble training via two steps: 1) relax the discrete random variable $k \in \{1, 2, \ldots, K\}$ into a continuous latent variable $c \in \mathbb{R}^L$ with a learned distribution that adaptively captures the diversity

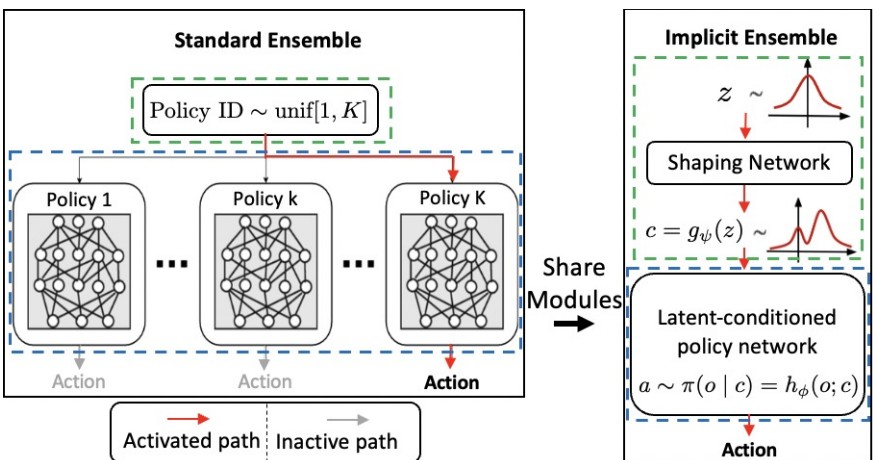

Figure 1: Connections between a standard ensemble and our proposed implicit ensemble: In the standard ensemble, a random index is sampled and the corresponding network is selected to output an action; while in our implicit ensemble (right), a random Gaussian variable is sampled and encoded into a latent vector by a shaping network, which is then passed to a conditional policy network to output an action.

of the policy ensemble; 2) replace the $K$ independent neural networks with a unified modular network architecture with parameter $\phi$ that improves parameter-efficiency by knowledge sharing between sub-modules within the network. The continuous relaxation also makes the policy differentiable with respect to the latent variable, thus making it possible to synthesize new policies from the learned skills represented by the sub-modules within the modular network by perturbing the latent variable distribution.

The generative process of this implicit ensemble is

$$
\begin{aligned}
z &\sim \mathcal{N}(\mathbf{0}, \mathbb{I}_{L \times L}), \\
c|z &= g_\psi(z), \\
a_i|c &\sim h_\phi(o_i; c).
\end{aligned}
\tag{5}
$$

In Eq. 5, a random Gaussian noise vector $z$ is sampled from the standard multivariate Gaussian distribution (sampled once at the beginning of each episode and remains fixed during the episode), then passed through a shaping network parameterized by $\psi$ to output a latent condition variable $c$. Finally, the action is sampled from a policy that is parameterized by $\phi$ and conditioned on the latent condition variable $c$. Both $\psi$ and $\phi$ are learnable parameters that are optimized end-to-end with respect to the reinforcement learning objective Eq. 2.

The implicit ensemble Eq. 5 is a more flexible parameterization than the ensemble training Eq. 3. In fact, the latter is a special case of the former, where the distribution of the latent variable $c$ collapses into a discrete distribution (corresponding to the uniform discrete distribution of $k$), and the shared parameter $\phi$ is divided into $K$ disjoint sets of parameters each of which is activated in the policy network when only one of the discrete values of $c$ is drawn.

In contrast to the ensemble training formulation Eq. 3, where the ensemble size $K$ is a hyperparameter to control the diversity of the (ensemble) policy, Eq. 5 does not contain this explicit hyperparameter. Instead, the diversity of the policy is captured adaptively by the learned latent distribution parameterized by $\psi$: a complicated multi-modal distribution corresponds to high diversity, while a simple uni-modal distribution corresponds to low diversity.

### 3.3 Model architecture

Fig. 1 shows the model architecture for implementing the IET. There are two major components within this architecture: 1) a shaping network, corresponding to the mapping $g_\psi(\cdot)$ in Eq. 5 that transforms the Gaussian

noise vector $z$ into the latent condition variable $c$; and 2) a conditional policy network, corresponding to the parametrization $h_\phi$.

The shaping network is responsible for adding diversity to the conditional policy for improving the robustness of the learned policy, and the conditional policy network improves parameter-efficiency via knowledge sharing. The detailed design of these two networks is presented in the following sections.

### 3.3.1 Shaping network

The shaping network takes a Gaussian noise vector $z \in \mathbb{R}^L$ as input and transforms $z$ into a latent condition variable $c$, which modifies the standard multi-variate Gaussian distribution into a learned (complicated) distribution. We use a feedforward network with 2 fully-connected layers followed by a Softmax layer to parameterize this shaping network.

### 3.3.2 Multi-tasking network

Recent studies (Yang et al., 2020; Devin et al., 2017; Huang et al., 2020) found that modular architecture is a parameter-efficient way of learning multi-tasking policies. In ensemble training, the multi-tasking requirement naturally arises from the fact that each policy is optimized against the ensemble of policies of the other agents.

To improve the parameter-efficiency of ensemble training, we use the modular architecture proposed in (Yang et al., 2020) as our conditional policy network for multi-tasking. We describe the detail of this modular network in the Appendix section. However, this specific network architecture is not the only choice for effective knowledge sharing. We show in Section 4.6 that a commonly used multi-head network is also effective for parameter-efficient ensemble training.

### 3.4 Model training

Algorithm 1 shows the pseudocode of the IET. At the beginning of each environment episode, each agent samples its own latent vector $z_i$ and then executes its conditional policy till the end of the episode. The latent vectors are fixed within each episode so that agents' policies are consistent across time steps. The model parameters $\psi_i$ and $\phi_i$ are trained through policy gradient update, which maximizes the long-term discounted cumulative reward objective for each agent.

---

**Algorithm 1** Implicit Ensemble Training

---

**Require:** Number of training episodes $M$, a set of agents with index in $\{1, \ldots, N\}$
  1: Initialize network parameters $\psi_i$, $\phi_i$, $i \in \{1, \ldots, N\}$ for each agent
  2: **for** $j = 1 : M$ **do**
  3:     **for** $i = 1 : N$ **do**
  4:         Sample latent vector $z_i \sim \mathcal{N}(\mathbf{0}, \mathbb{I}_{L \times L})$
  5:     **end for**
  6:     Rollout with each agent sampling its action via $a_i \sim h_{\phi_i}(o_i; g_\psi(z_i))$
  7:     Update network parameters $\psi_i$ $\phi_i$ with policy gradient
  8: **end for**
  9: **return** $\psi_i$, $\phi_i$

---

Although Algorithm 1 shows that each agent independently samples its private latent vector, it is beneficial to share the latent vectors between agents under certain situations. For example, consider a mixed cooperative-competitive scenario where two teams of agents compete against each other. Agents within the same team are incentivized to communicate their latent vectors or condition their policies on the same latent vector (which could be achieved without communication by using a common random seed) to achieve effective coordination. However, since the focus of this paper is on policy robustness, later sections of this work focus on experimental evaluation of the proposed IET approach in competitive scenarios involving two agents.

## 4 Evaluation

### 4.1 Scenarios

We evaluate our approach on two types of 2-player (which we refer to as the blue agent and the red agent hereafter) multiagent scenarios.

- **Board-game**: turn-based games implemented in the PettingZoo multiagent environment[2] (Terry et al., 2020) and the RLCard toolkit[3] (Zha et al., 2019): **Connect Four**, **Leduc Hold'em**, and **Texas Hold'em (Limit)**.

- **RoboSchool-Racer**: continuous problems modified from the robot racing scenarios in the RoboSchool environment[4]: **Ant** and **Cheetah**, where we decompose each robot into front and rear parts, and assign opposite rewards to each part. As such, the front is learning to move forward while the rear is learning to move backward.

### 4.2 Baselines

We compare our approach with the following baselines:

1. **Single Policy Training** (SPT): a standard multiagent training approach wherein each agent optimizes only one policy.

2. **Simple Ensemble Training** (SET): a standard ensemble training approach wherein each agent optimizes an ensemble of policies against each other. We use an ensemble size of 3 for each agent (an ensemble of 3 policies has been shown to improve robustness significantly in previous works (Lowe et al., 2017) and (Bansal et al., 2017)), which also increases the amount of computation by a factor of 3.

3. **Minimax Training** (MT): achieves worst-case robustness through optimizing one's policy against the worst-case opponents, which is formulated as a minimax optimization. We use the one-step approximation approach in (Li et al., 2019) with the Multiagent Actor-attention-critic (MAAC) algorithm (Iqbal & Sha, 2019)

### 4.3 Implementation detail

We used the RLlib (Liang et al., 2018) implementation of Proximal Policy Optimization (PPO) with a mini-batch size of 256 and a learning rate of $5 \times 10^{-5}$. We use independent networks for the policy and the value function approximations and set the following hyperparameters for IET: $L = 10$ for the latent condition variable dimension; $H = 64$ for the hidden layer dimension in the shaping network; $n = 2$ and $m = 2$ for the number of layers and number of modules of the modular network; $D = 64$ and $d = 64$ for the embedding and module hidden dimension. For the other approaches, each of the policy and the value networks consists of two fully-connected layers with 256 hidden units.

In the simple ensemble training setting, each sub-policy within an ensemble only has a probability of 1/3 to be selected for execution. If the same number of environment rollouts is used to train the simple ensemble as used for the other training settings, the simple ensemble policy will perform poorly due to insufficient training. Therefore, we roll out two additional environment simulations (but counted as one training step when training the simple ensemble) for a fair comparison at the cost of additional computational overhead.

To evaluate the robustness of the learned policies, we adopted a similar approach as in (Vinyals et al., 2019; Gleave et al., 2019) by training an independent exploiter agent. Specifically, we launched two concurrent threads, one for the training, the other for the testing, and repeated the following steps:

---

[2] https://www.pettingzoo.ml
[3] https://github.com/datamllab/rlcard
[4] https://github.com/openai/roboschool

1. Train the blue agent and the red agent in the training thread for one training epoch.

2. Copy the blue agent's policy to the testing thread and freeze it.

3. Train the red exploiter agent in the testing thread against the fixed blue agent.

As a result, the red exploiter agent learns to exploit any weakness of the blue policy, and the corresponding reward is an informative indicator of the adversarial robustness of the blue policy.

### 4.4 Results and comparisons

This section discusses experimental results and compares baseline approaches with our proposed IET approach.

#### 4.4.1 Adversarial robustness

We investigate the adversarial robustness of the learned blue policy by training a red exploiter agent against the frozen blue agent policy.

Table 1 shows the average testing reward, the best testing reward, and the average reward gap between training and testing. We compare our IET approach with two baselines. One baseline is our IET approach with the input latent noise fixed as a zero vector, and the other baseline is a standard ensemble with 10 policies. For the IET settings, we ran experiments across 25 random seeds. However, we found that due to the difficulty of balancing the relative strength between the blue and the red agents, sometimes (for some random seeds) the blue agent's training reward converges to near -1.0, and so does the testing reward. Since we are interested in investigating the robustness of the learned policy (whether a strong training performance is sufficient to guarantee a strong testing performance), we filter out those seeds that lead to poor training performance (converged training reward lower than -0.5). There turned out to be 19 remaining valid seeds for each of the two IET settings, and the results are evaluated only across these valid seeds. As for the SET setting, we found that both the training and the testing reward are not very sensitive to random seeds, so we report the results across 4 random seeds. Our IET approach outperforms the baselines for both the mean testing reward and the best testing reward, especially for the besting testing reward where our IET reached near optimal testing reward within 4 out of the 19 valid seeds.

Table 1: Testing reward against exploiter and reward degradation

| Connect Four | IET (fixed latent) | SET (10) | IET |
|---|---|---|---|
| **Mean testing reward** | -0.40 $\pm$ 0.08 | -0.21 $\pm$ 0.06 | **-0.06 $\pm$ 0.14** |
| **Best testing reward** | -0.16 | -0.03 | **1.0** |
| **Mean training/testing reward gap** | 0.70 $\pm$ 0.18 | **0.17 $\pm$ 0.08** | 0.42 $\pm$ 0.14 |

### 4.5 Competition scores between training settings

To evaluate the out-of-distribution robustness, we also include the competition scores between the blue agent and the red exploiter agent, calculated as $(r_{\text{blue}} - r_{\text{red}})/2$ for each pair of training settings in Tables 2. The blue agent uses the policy learned in the training thread, while the red agent uses the exploiter policy learned in the testing thread. As a result, the diagonal scores are in favor of the red agent, because the red exploiter policy is trained against the corresponding blue policy, but not the other way around. The diagonal scores measure the adversarial robustness of the learned blue policy. In contrast, the off-diagonal scores measure the out-of-distribution robustness of the blue policy, because neither the blue policy nor the red policy has been trained against each other. However, drawing conclusions directly from Tables 2 about the relative strength of the policies learned by the three training settings is difficult, because the competition scores show that there is no single policy that can dominate all the rest policies. This non-transitive nature of real-world games has also been observed in previous work (Czarnecki et al., 2020).

Table 2: Competition scores between SPT, SET, and IET. Higher score implies that the blue policy is stronger than the red policy. The last column lists the lowest scores over the columns for each row, where our IET achieves the highest scores among the training settings.

| Connect Four | SPT | SET | IET | Lowest across columns |
|---|---|---|---|---|
| SPT | -1.0±0.0 | -0.16±0.04 | 0.29±0.04 | -1.0±0.0 |
| SET | 0.35±0.04 | -0.98±0.01 | 0.65±0.03 | -0.98±0.01 |
| IET | 1.0±0.0 | 1.0±0.0 | -0.33±0.04 | **-0.33±0.04** |

| Leduc Hold'em | SPT | SET | IET | Lowest across columns |
|---|---|---|---|---|
| SPT | -0.63±0.12 | -0.34±0.09 | 0.12±0.07 | -0.63±0.12 |
| SET | 0.55±0.14 | -0.04±0.1 | -0.07±0.09 | -0.07±0.09 |
| IET | 0.47±0.12 | 0.11±0.09 | -0.03±0.09 | **-0.03±0.09** |

| Texas Hold'em | SPT | SET | IET | Lowest across columns |
|---|---|---|---|---|
| SPT | -1.47±0.32 | -0.12±0.11 | 0.0±0.05 | -1.47±0.32 |
| SET | -0.6±0.27 | -0.16±0.18 | 0.05±0.10 | -0.6±0.27 |
| IET | 0.39±0.24 | -0.09±0.16 | 0.13±0.12 | **-0.09±0.16** |

To quantitatively evaluate the robustness of the learned blue policy, we provide two metrics: (1) Normalized score: we find the worst-case reward gap between blue and red $\min_{\pi_r \in \Pi}[r_{\text{blue}}(\pi_b, \pi_r) - r_{\text{red}}(\pi_b, \pi_r)]$, $\Pi = [\text{SPT}, \text{SET}, \text{IET}]$ (higher is better) and normalize it to $[0, 1]$, (2) Nash policy probability: we follow the approach proposed in (Balduzzi et al., 2018) by solving for the Nash-Equilibrium (NE) (Holt & Roth, 2004) of the meta-game (Tuyls et al., 2020), which involves a row-player and a column-player, whose actions are selecting which row/column policy to execute from the three available policies. The NE of the meta-game is a pair of stochastic policies. The probability of a policy being selected by the meta-player measures the strength (robustness) of this policy.

We show these two metrics in Table 3 and Table 4. The results show that our IET approach consistently outperforms the other ones across all the scenarios.

Table 3: Normalized scores that measure the adversarial robustness.

| Scenarios / Settings | SPT | SET | IET |
|---|---|---|---|
| Connect Four | 5e-3 | 1e-2 | **0.33** |
| Leduc Hold'em | 0.18 | 0.46 | **0.48** |
| Texas Hold'em | 0.0 | 0.2 | **0.46** |

Table 4: Nash policy probability of the meta-player that measures the overall strength of the policy.

| Scenarios / Settings | SPT | SET | IET |
|---|---|---|---|
| Connect Four | 0.0 | 0.38 | **0.62** |
| Leduc Hold'em | 0.0 | 0.41 | **0.59** |
| Texas Hold'em | 0.0 | 0.0 | **1.0** |

### 4.5.1 Comparison with the minimax robust approach

We also show in Table 5 the comparison between the ensemble training approaches with the minimax robust learning approach (Li et al., 2019) which maximizes the worst-case reward to obtain adversarial robustness. We use the RoboSchool environment because the minimax approach requires differentiability with respect to the action (therefore continuous action environments).

As the complexity of the environments increases, we see that the scores become noisier due to the difficulty of the reinforcement learning algorithm to optimize the policy (indicated by the fact that fewer minimal scores across the column are achieved at the diagonal). Again, we show the Nash meta-policy probability in

Table 5: Competition scores between SPT, SET, IET, and MT. IET achieves the best adversarial robustness in **Ant** as well as best out-of-distribution robustness in both **Ant** and **Cheetah**.

| Ant | SPT | SET | IET | MT | Lowest across columns |
|---|---|---|---|---|---|
| SPT | 8.42±0.26 | 26.2±0.82 | 1.95±0.54 | -4.49±2.86 | -4.49±2.86 |
| SET | -35.3±1.54 | 5.09±1.0 | -23.7±0.86 | -24.5±2.03 | -35.3±1.54 |
| IET | 4.13±0.84 | 33.64±1.03 | 15.2±0.38 | 37.7±0.81 | **4.13±0.84** |
| MT | -25.37±1.73 | -2.53±0.90 | -13.85±1.44 | -25.7±0.50 | -25.7±0.50 |

| Cheetah | SPT | SET | IET | MT | Lowest across columns |
|---|---|---|---|---|---|
| SPT | -15.54±0.68 | -15.95±0.72 | -7.16±0.34 | 4.26±0.22 | -15.95±0.72 |
| SET | -9.05±0.21 | -12.60±0.64 | -1.76±0.41 | 6.45±0.29 | -12.60±0.64 |
| IET | -9.22±0.22 | -11.88±0.27 | 0.24±0.37 | 8.16±0.24 | **-11.88±0.27** |
| MT | -13.71±0.59 | -20.12±0.65 | -4.02±0.50 | 4.84±0.38 | -20.12±0.65 |

Table 6, which suggests our IET approach achieves the best robustness. The fact that the minimax robust approach does not work very well could be a result of two reasons: 1) The minimax formulation may not be a good approach to achieve out-of-distribution robustness since it optimizes for the worst-case reward, which may lead to overly-conservative behaviors that fail to exploit the weaknesses of the opponent; 2) The one-step solution technique for the minimax problem proposed in (Li et al., 2019) is an approximate solution, which may find a sub-optimal solution due to the difficulty of selecting a suitable step-size parameter, which has been reported in previous work (Sun et al., 2021).

Table 6: Nash meta-policy probability in the RoboSchool scenarios.

| Scenarios / Settings | SPT | SET | IET | MT |
|---|---|---|---|---|
| **Ant** | 0.0 | 0.37 | **0.63** | 0.0 |
| **Cheetah** | 0.0 | 0.0 | **1.0** | 0.0 |

### 4.6 Ablation studies

To further understand the role of the shaping network and the multi-tasking network, we conduct two additional experiments: 1) We show that the shaping network learns a non-trivial mapping from the Gaussian noise to the latent variable only when diversity is required through visualizing the mappings represented by the shaping networks learned in multiagent scenarios and compare those with that learned in a single-agent scenario; 2) We show that we can improve the sample-efficiency of the PSRO algorithm by sharing the intermediate layers within a multi-tasking network, and the optimal performance is obtained at a medium level of sharing.

#### 4.6.1 Policy diversity

To verify our conjecture that the robustness of our IET approach is a result of IET's capability to present a diverse set of policies, we show the histogram of the action probabilities (Fig. 3) in the Connect Four game, and the corresponding training and testing rewards (Fig. 2). Fig. 2a shows that the IET agent's testing reward increases significantly at around 60 million training steps. Fig. 3a and 3b show the action probability histograms of the IET agent at 15 million training steps, which is almost uni-modal. This lack of diversity explains the poor testing reward during the early training stage (0-60M steps). In contrast, Fig. 3c and 3d shows the histograms at 60 million training steps, which span a much wider range of action probabilities. Specifically, Fig. 3c corresponds to the first time step of the game only, where the agent observation is always empty board (deterministic). Therefore, the wide range of action probabilities is a result of the policy conditioning on the latent noise. Fig. 3e and 3f shows the histogram at 75 million training steps, which is also uni-modal. Notice that the corresponding testing reward shown in Fig. 2 is near optimal. This result suggests that the IET agent has learned a strong policy (Another evidence is that the probability of taking action 3, which corresponds to placing the first chip in the middle of the board, is near 1.0. It is known that the optimal strategy in Connect Four for the first player is to place the chip in the middle.),

which is in accordance with the fact that the optimal policy in Connect Four is a deterministic policy. Our results are also in accordance with the findings in (Czarnecki et al., 2020) that this Connect Four game has a spinning top structure: when the policy is very weak (e.g. at 15M steps) or very strong (e.g. at 75M steps), there is little diversity; but within the intermediate region of the policy strength spectrum (e.g. at 60M steps), there are many diverse policies; training against a sufficiently diverse set of policies will lead to improved policy strength.

For comparison, Fig. 3g and Fig. 3h show the histograms of a SET agent of ensemble size 10, where the SET agent's action probabilities only span a few discrete values, implying less diversity as compared against the IET agent at 60 million training steps. These results explain the superior testing reward achieved by the IET agent.

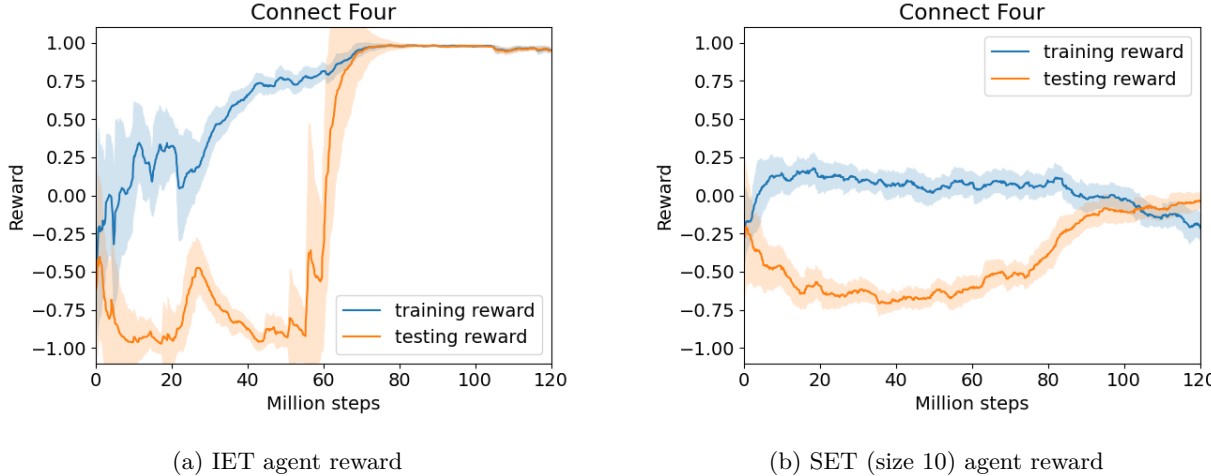

(a) IET agent reward             (b) SET (size 10) agent reward

Figure 2: Training and testing rewards of IET agent (left) and SET agent of ensemble size 10 in the Connect Four scenario.

### 4.6.2 Ablation on the levels of sharing

We investigate how the level of sharing within the multi-tasking network influences the sample-efficiency. As our approach is not restricted to the modular network architecture, for the convenience of ablation, we instead use a more intuitive multi-head multi-tasking network architecture consisting of $L = 5$ fully connected layers, where the first $L_{\text{sharing}}$ layers are shared and the last $L - L_{\text{sharing}}$ layers are independent for each policy. We show in Fig. 4a, 4b, 4c and 4d the exploitability and its area under the curve (AUC) of the joint policy when running the PSRO algorithm with the multi-tasking network of different sharing levels. With $L_{\text{sharing}} = 0$ (independent policies) corresponds to the standard PSRO, and $L_{\text{sharing}} = 5$ (identical policies) corresponds to self-play. We see that the best exploitability descent happens at $L_{\text{sharing}} = 2$, while the two extremes ($L_{\text{sharing}} = 0, 5$) perform poorly. This observation suggests a trade-off between knowledge sharing (positive transfer) and loss of flexibility (negative transfer), which is commonly observed in multi-task learning. This ablation study also verifies our design purpose that the multi-tasking network in our implicit ensemble approach is responsible for improving the sample-efficiency via sharing parameters.

Table 7 shows the scaling of network parameters and the representational power (policy diversity) of different approaches to parameterize the policy ensemble, where $K^{\text{policy-net}}$ denotes the number of parameters of the policy network, $K^{\text{head}}$ and $K^{\text{shared-net}}$ denote the number of parameters of the policy head and shared network in the multi-head multi-task network, and $N$ denotes the number of policies within the ensemble. For the standard simple ensemble approach that uses independent networks for each policy, the total number of parameters scales linearly w.r.t. the number of policy $N$. For the multi-head ensemble that uses independent heads with a shared base network to parameterize policies, the number of parameters also scales linearly, where the scaling coefficient is $K^{\text{head}}$. For a small $K^{\text{head}}$, the parameter complexity is low. However, in this case, since the shared network will become the dominant part of this multi-head network, the diversity of the

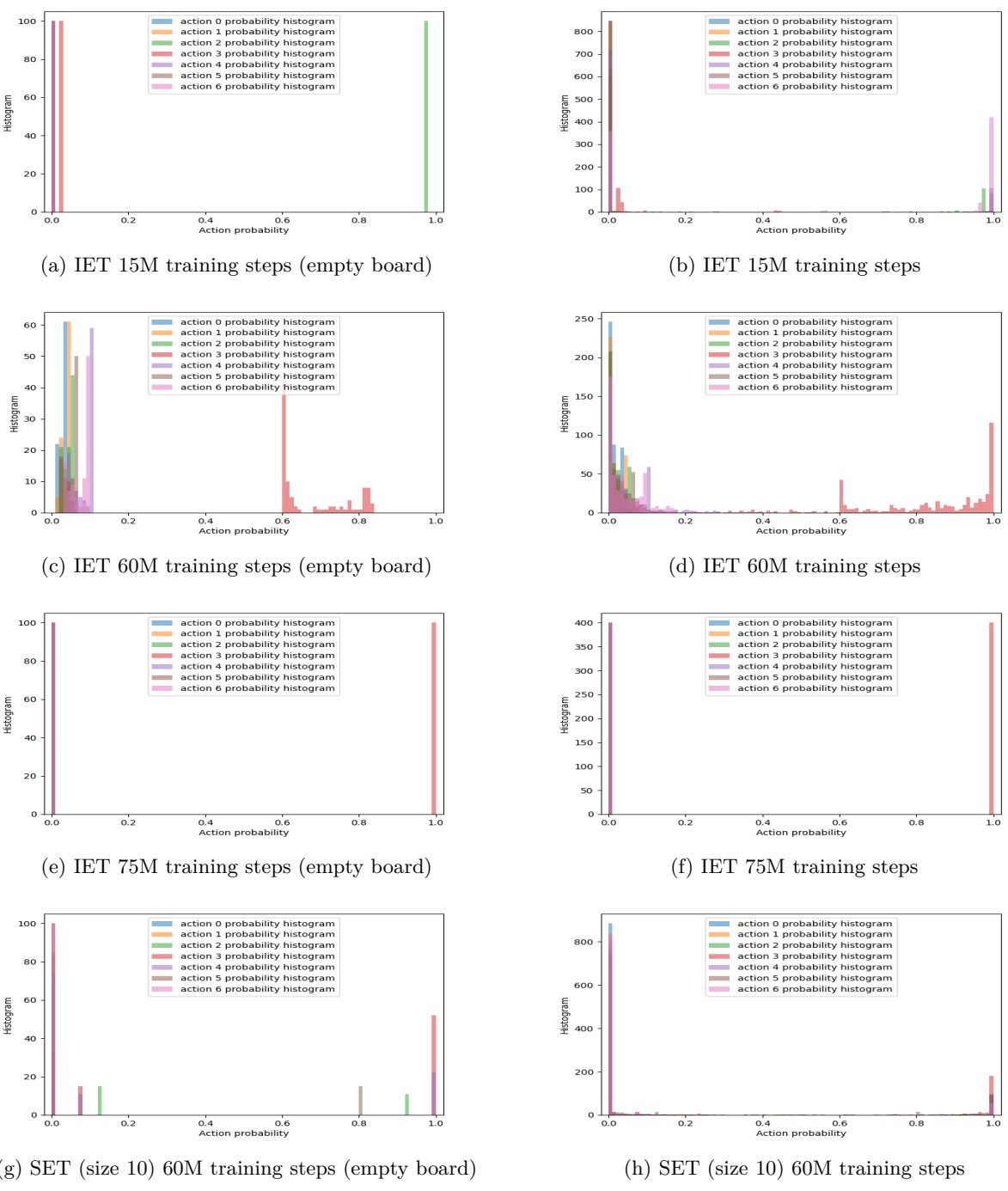

Figure 3: Histogram of action probabilities corresponding to the rewards shown in Fig. 2 at snapshots of the IET agent (at 15, 60, 75 million training steps) and the SET agent of ensemble size 10 (at 60 million training steps): 3a, 3c, 3e and 3g show the histogram of action probabilities at the first time step of the game (empty board as in Connect Four); 3b, 3d, 3f and 3h show the histogram of action probabilities accumulated across all the time steps of the game. IET agent's policy at 60M training steps covers a wide range of action probabilities.

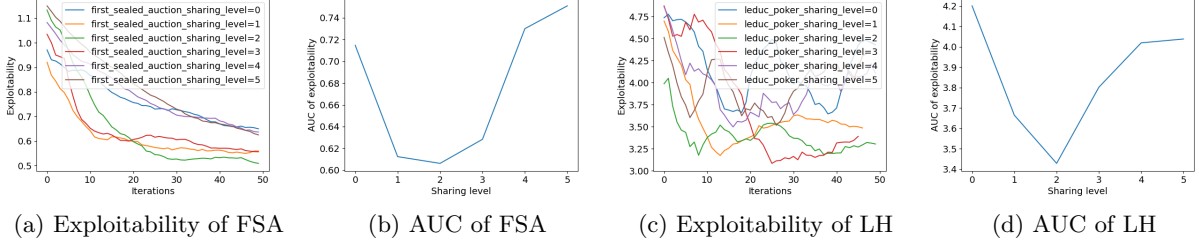

(a) Exploitability of FSA      (b) AUC of FSA      (c) Exploitability of LH      (d) AUC of LH

Figure 4: Exploitability (interpreted as a distance from Nash Equilibrium (Lanctot et al., 2017)) and the area under the curve (AUC) of the joint policies learned through running PSRO with the multi-task network of different sharing levels. The optimal exploitability descent happens at medium sharing levels ($L_{\text{sharing}} = 2$). Scenarios First Sealed Auction (FSA) and Leduc Hold'em (LH) implementation from OpenSpiel (https://github.com/deepmind/open_spiel).

policy ensemble parameterized by this multi-head network is compromised. In contrast, for a large $K^{\text{head}}$, the policy diversity is improved, but the parameter complexity scales poorly. For the implicit ensemble that uses a single shared network with a latent random variable to parameterize policy distribution, the parameter complexity is constant. The policy diversity arises from the randomness of the continuous latent variable. As a result, the implicit ensemble can represent a continuous distribution of policies.

Table 7: Parameter complexity and policy diversity corresponding to various approaches of parameterizing policy ensemble

| **Approaches** | Simple Ensemble | Multi-head Ensemble | Implicit Ensemble |
|---|---|---|---|
| Parameter complexity | $K^{\text{policy-net}}N$ (Linear) | $K^{\text{head}}N + K^{\text{shared-net}}$ (Linear) | $K^{\text{shared-net}}$ (Constant) |
| Policy diversity | $N$ distinct policies | $N$ distinct policies | Continuous distribution of policies |

## 5 Related works

This work is at the intersection of robust MARL and multi-task RL.

### 5.1 Robust MARL

Ensemble training for robust MARL has been studied in many previous works such as (Lowe et al., 2017; Bansal et al., 2017; Jaderberg et al., 2017; 2019; Vinyals et al., 2019). These approaches focus on improving the robustness of the learned policies, regardless of the associated increase of computational complexity. In contrast, our work focuses on improving the efficiency of ensemble training without sacrificing robustness. Another approach for robust MARL is minimax policy optimization, in which each agent optimizes its policy assuming all the other agents and the environment dynamics are adversarial (see (Zhang et al., 2020; Li et al., 2019)). One difficulty with this approach is the requirement of solving the nested minimax optimization, which is typically approximated by optimizing the inner minimization for one step only (Li et al., 2019). In addition, the minimax formulation tends to result in overly-conservative policies because of the pessimistic assumption about the other agents.

### 5.2 Multi-task Reinforcement learning

Our work is related to multi-task reinforcement learning (MTRL). MTRL aims to improve the learning efficiency by knowledge sharing across tasks (D'Eramo et al., 2019). Recent works (Huang et al., 2020; Yang et al., 2020; Devin et al., 2017) found that the modular policy network is an effective architecture for improving parameter-efficiency via sharing of learned modules. However, that MTRL work focuses on solving the single-agent multi-task problem. In contrast, our work leverages the multi-tasking network architecture combined with latent conditional policy learning (Haarnoja et al., 2018; Lee et al., 2019) to improve the efficiency of ensemble-based robust MARL. The innovation of our approach is re-formulating

ensemble training as multi-task learning while using a latent variable to preserve policy diversity which is essential for robust MARL.

### 5.3   Entropy-regularized Reinforcement Learning

At first glance, our approach has similarities to entropy regularized RL with Gaussian policy (Haarnoja et al., 2017; Wei et al., 2018), which has the benefit of improved exploration. The key difference is that entropy regularized RL samples a random Gaussian noise at each time step, while our approach samples a policy from a continuous policy space by sampling a latent Gaussian noise at the beginning of each episode and keeps it fixed across the whole episode. This difference is analogous to the difference between a stochastic behavioral strategy and a mixed strategy (Walker & Wooders, 2006) in game theory. Although adding randomness at each time step is helpful for smoothing the RL learning objective to avoid local minimum (Ahmed et al., 2019), it is not as effective as randomizing over the policy for creating diversity, which enables our IET approach to improve policy robustness.

### 5.4   Stochastic Neural Network

Stochastic neural networks such as networks with dropout layers (Gal & Ghahramani, 2016) and Osband et al. (2016) is another effcient way of parameterizing policy distribution. Compared with these approaches, our IET parametrization is more flexible and expressive. The dropout approach in (Gal & Ghahramani, 2016), for example, involves the dropout rate as a hyper-parameter, which determines the diversity of the policy ensemble. As we discussed earlier, the required policy diversity for learning robust policy distribution in real-world games also evolves with the training stage and policy strength, which makes it difficult to select a fixed dropout rate beforehand. In contrast, our approach enables adaptive policy diversity that is arguably more suitable for learning in these settings.

## 6   Conclusions

This paper proposes IET, an implicit ensemble training approach that effectively reduces the computational complexity of ensemble training while maintaining the policy diversity for learning robust multiagent policies. In contrast to previous ensemble training approaches that require optimizing multiple policy networks, our IET approach optimizes a single shared network, which requires less computation and memory. Numerical results show that our approach improves both the learning efficiency and the robustness of the learned policy.

We identify two promising future directions to extend our work: 1) Improving the stability of our IET training. Our current approach does not impose any regularization on the learned representation of the latent conditional vector, which sometimes leads to mode collapse. A mutual information regularizer might be helpful to avoid mode collapse. 2) Our approach can be straightforwardly extend to scenarios where two teams of agents compete with each other. The latent vector can be shared across agents within the same team which acts as a shared signal for team coordination to naturally emerge.

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

## A   Appendix A

We use the soft modular network proposed in (Yang et al., 2020) as the multi-tasking sub-network in our IET approach. The modular network is composed of a base network and a routing network. The base network is a modular network that has $n$ layers, and each layer has $m$ modules. Each module is a feedforward network, and the input and the output are $d$-dimension vectors. The routing network takes the observation and the latent condition variable as inputs, and it then outputs the $n$ normalized weight vectors, one for each layer, for weighting the modules of the base policy network. The weight vectors $w^j \in \mathbb{R}^{m^2}$ are calculated as

$$
\begin{aligned}
p^{j=1} &= W_d^{j=1} \left( \sigma \left( F\left(o\right) \cdot H\left(c\right) \right) \right), \\
p^{j+1} &= W_d^j \left( \sigma \left( W_u^j p^j \cdot \left( F\left(o\right) \cdot H\left(c\right) \right) \right) \right), \\
w^j &= \text{Softmax}(p^j),
\end{aligned} \tag{6}
$$

where $F$ and $H$ are the embedding layers, which map the observation vector $o$ and the latent condition variable $c$ into $D$-dimension embeddings. $\sigma$ is the activation function (we use the ReLU activation). $W_u$ and $W_d$ are fully-connected layers of size $\mathbb{R}^{D \times m^2}$ and $\mathbb{R}^{m^2 \times D}$, respectively.

The base network takes the observation vector $o$ as input and outputs policy logits / value, with the following relationship between the $i$-th module in the $j{+}1$ layer's input and the $k$-th module in the $j$-th layer's output,

$$\hat{f}_i^{j+1} = \sum_{l=1}^{m} w_{i,l}^j \left( \sigma \left( W_l^j \hat{f}_l^j \right) \right), \tag{7}$$

where $W_l^j \in \mathbb{R}^{d \times d}$ is the learnable module parameters.

