# OpenReview forum: "Implicit Ensemble Training for Efficient and Robust Multiagent Reinforcement Learning"
_TMLR — Accepted by TMLR_

### Review · Reviewer_mj1G · 2022-08-01

**Summary Of Contributions:**

This paper studies the topic ensemble training, which is an effective approach for learning robust policies that avoid significant performance degradation when competing against previously unseen opponents, since a large ensemble can improve diversity during the training, which leads to more robust learning. This paper proposes a parameterization trick of a policy ensemble based on a deep latent variable model with a multi-task network architecture, and demonstrate in several competitive multiagent scenarios in the board game and robotic domains that our new approach improves robustness against unseen adversarial opponents while achieving higher sample-efficiency and less computation.

**Requested Changes:**

●	Improvements to the related work section as mentioned in the weaknesses.
●	Update figures 2,3,4 for improved clarity about what each line represents and make it clearer why these figures show good performance of IET.
●	Improvements to the baselines used as mentioned in the weaknesses.
●	Experiment looking at team settings in mixed competitive cooperative environments.


**Strengths And Weaknesses:**

**Strengths**

●	A novel solution to a real problem in ensemble methods is proposed. By proposing a single model that represents different agents dependent on the sampled latent variable, they are able to mitigate the major issue of computation scaling linearly with individual models for each ensemble agent.

●	Much of the paper is well-written and also well-motivated.

●	There has been a good initial attempt at the evaluation. For example, I believe the selection of environments is reasonably complex and I found the ablation studies, in particular on the levels of sharing, to be thought-provoking for the usage of single-models in ensemble methods.

**Weaknesses**

●	In section 3.4 there is discussion on how the latent vector can be used in the mixed cooperative-competitive setting in which two teams of agents compete against each other. However, there is no example of this in the evaluation section. In my opinion, if it could be shown that the IET method is able to produce effective coordination between agents in a team then this would be greatly increase the benefit and appeal provided by IET.

●	There are aspects of the conditional variable that are unclear to me. Mainly, the sampling of the latent vector and the outputted latent condition variable do not seem to have any relevance to the observations of the environment nor the opponent that is being trained against. Understandably, I may not be fully grasping what is happening - but my instinct would be that the conditioning variable that essentially dictates how the agent is going to play in the environment should be influenced by the opponent in some way. Please could you clarify where I may be incorrect, or how the conditioning variable is influenced by the opponent?

●	In my opinion, the related work and how this work fits in misses two critical areas.

●	Firstly, the discussion with respect to PSRO methods is limited and does not mention any examples of extensions to PSRO that aim to deal with scalability problems. For example - P2SRO [McAleer et al., 2020], NAC [Feng et al., 2021]. In addition, I view PSRO style methods as the major competitor to this approach in terms of performance and would expect there to, in general, to have been a wider discussion to PSRO methods that are not also just focused on improved scalability - Rectified PSRO [Balduzzi et al., 2019], Diverse PSRO [Nieves et al., 2021] JPSRO [Marris et al., 2021]

●	Secondly, NeuPL [Liu et al., 2022] was introduced recently and I am surprised this has not been discussed by this work. NeuPL additionally utilises a single conditional model that conditions on the strategy of the opponent in order to improve the transfer of skills between agents. It would be great to hear the authors thoughts on how IET improves over NeuPL and also maybe falls short of NeuPL.

●	In my opinion, the evaluation section is not extensive enough in terms of baselines.

●	Firstly, a similar weakness to the one above but I think the major competitor to IET is PSRO methods. It would be great to see an environment where the amount of PSRO policies needed becomes intractable, whilst IET is able to handle the problem due to its ability to represent a larger amount of policies more efficiently.

●	Similarly again, it would be great to see a comparison to NeuPL if feasible.

●	In terms of the Simple Ensemble Training, the setup seems rather basic. For example, an ensemble size of 3 feels small in comparison to the amount of diversity that I believe IET can show, but also why is uniform sampling over the ensemble utilised? Is not using something like the Nash distribution over the policies more standard, and potentially a fairer comparison?

●	Personally, I found figures 2,3,4 confusing to understand most notably due to their poor legends.

---

> ### Author Response · Authors · 2022-08-10
> **Responses**
>
> Thanks for the comments. Here are our responses:
>
> ```● In section 3.4 there is discussion on how the latent vector can be used in the mixed cooperative-competitive setting in which two teams of agents compete against each other. However, there is no example of this in the evaluation section. In my opinion, if it could be shown that the IET method is able to produce effective coordination between agents in a team then this would be greatly increase the benefit and appeal provided by IET.```
>
> Thanks for your suggestion on adding experiments of IET for effective coordination. We conjecture that the extension of our IET approach to mixed cooperative-competitive scenarios could be related to the correlated equilibrium concept in game theory, where each player chooses their action according to their private observation of the same public signal. In our approach, the random noise serves as such public signal (public only within the same team, but not with the opponent team). That being said, we have not run experiments to verify this conjecture yet. Due to the limited computational resource we have and the higher requirement of running such experiments, we will only try it if given enough time beyond the review period, or we may leave it for future work.
>
> ```● There are aspects of the conditional variable that are unclear to me. Mainly, the sampling of the latent vector and the outputted latent condition variable do not seem to have any relevance to the observations of the environment nor the opponent that is being trained against. Understandably, I may not be fully grasping what is happening - but my instinct would be that the conditioning variable that essentially dictates how the agent is going to play in the environment should be influenced by the opponent in some way. Please could you clarify where I may be incorrect, or how the conditioning variable is influenced by the opponent?```
>
> The conditioning variable is influenced by the opponent as such: Since the shaping network that determines the distribution of the conditioning variable is trained to optimize the cumulative reward subject to the transition dynamics dictated by both agents' policies, the opponent policy will influence the learning of this shaping network, and as a result influence the distribution of the conditioning variable. Another way to understand this influence could be making analogy betwwen our IET approach and mixed-strategy equilibrium: the multi-tasking network may be interpreted as outputting a set of pure strategies, while the conditioning variable is controlling the mixture coefficients corresponding to the pure strategies (another justification of using softmax activation Eq. 6 in the Appendix in the intermediate layers). So if we freeze the multi-tasking network and only train the shaping network, it could be interpreted as solving for the meta-game over a fixed set of pure strategies. The meta-strategy dictated by the distribution of the conditioning variable is clearly influenced by the opponent's meta-strategy. But since we do not freeze the multi-tasking network, there is no clear separation between updating the pure strategies and the meta-strategy. Instead, these two interleaving processes happen end to end, which may obscure the role of each components.

---

> > ### Author Response · Authors · 2022-08-15
> > **Responses Continued**
> >
> > ```● In my opinion, the related work and how this work fits in misses two critical areas.```
> >
> > ```● Firstly, the discussion with respect to PSRO methods is limited and does not mention any examples of extensions to PSRO that aim to deal with scalability problems. For example - P2SRO [McAleer et al., 2020], NAC [Feng et al., 2021]. In addition, I view PSRO style methods as the major competitor to this approach in terms of performance and would expect there to, in general, to have been a wider discussion to PSRO methods that are not also just focused on improved scalability - Rectified PSRO [Balduzzi et al., 2019], Diverse PSRO [Nieves et al., 2021] JPSRO [Marris et al., 2021]```
> >
> > Thanks for pointing us to the extensions of PSRO and NeuRL. We have attempted to compare our approach against the PSRO family of works, but with little success because of two reasons: (1) Feasibility: For example, Diverse PSRO is proposed based on normal-form games, where the strategies have tabular form. It is unclear to us how to evaluate and optimize the diversity metric derived from the determinantal point process on deep policies parameterized by a neural network in a computationally tractable way; (2) Fairness: The PSRO family of approaches uses a growing set of policies (so the computation and memory complexity is growing too), which makes it difficult to come up with a fair comparison with the ensemble based approaches which has constant computation and memory complexity. Also, the PSRO family of approaches require solving a meta-game explicitly each iteration. As the number of policies increases, the complexity of finding the meta-game payoff matrix may scale quadratically. This becomes problematic when the game itself has high randomness (e.g. texas hold'em) or stochastic policies are used, because it may need a large number of games to be played to reduce the variance of each entry in the payoff matrix. We would appreciate it if the review has insights on how to make a fair comparison.
> >
> > ```● Secondly, NeuPL [Liu et al., 2022] was introduced recently and I am surprised this has not been discussed by this work. NeuPL additionally utilises a single conditional model that conditions on the strategy of the opponent in order to improve the transfer of skills between agents. It would be great to hear the authors thoughts on how IET improves over NeuPL and also maybe falls short of NeuPL.```
> >
> > ```● In my opinion, the evaluation section is not extensive enough in terms of baselines.```
> >
> > ```● Firstly, a similar weakness to the one above but I think the major competitor to IET is PSRO methods. It would be great to see an environment where the amount of PSRO policies needed becomes intractable, whilst IET is able to handle the problem due to its ability to represent a larger amount of policies more efficiently.```
> >
> > ```● Similarly again, it would be great to see a comparison to NeuPL if feasible.```
> >
> > We would like to thank the reviewer again for pointing us to NeuPL, which is an important related work that we have missed. Our current submission is a direct extension to a workshop paper that we submitted in 2021, which was published before the NeuPL paper. We should have redone a literature review before writing this extension so that we would not miss this related work. After reading NeuPL, we see that this work indeed shares a few similar ideas as our IET does (conditioning variable and shared network policies). So we believe it would be beneficial to focus on discussing the difference between these two approaches. (1) NeuPL adopts a similar approach as PSRO in the sense that it explicitly parameterizes a finite set of policies, and formulates a meta-game at each iteration to derive a new policy. In contrast, our IET approach implicitly represents an infinite set of policies (continuous mixture), without explicitly formulating a meta-game. Instead, as alluded above, there is an implicit meta-game played between the two agents, through end to end training of the shaping network along with the multi-tasking network. (2) Compared against PSRO, NeuPL has the benefit of finding adaptive number of policies as required by the nature of the game. In fact, our IET approach also has a similar benefit because the shaping network could have learned a Dirac delta measure for the conditioning variable, or the multi-tasking network could have learn to neglect the conditioning variable, depending on the transitivity nature of the game. (3) NeuPL formulates a meta-game which explicitly creates a new policy that is distinct than all the old policies. However, in our IET, the only driven force for diverse policies is the nature of the game and the reward maximization objective. We will add a separate paragraph in the related work section for the discussion of NeuPL, and try to add an experiment comparison with NeuPL if time permits.

---

> > > ### Author Response · Authors · 2022-08-15
> > > **Responses Continued**
> > >
> > > ```● In terms of the Simple Ensemble Training, the setup seems rather basic. For example, an ensemble size of 3 feels small in comparison to the amount of diversity that I believe IET can show, but also why is uniform sampling over the ensemble utilised? Is not using something like the Nash distribution over the policies more standard, and potentially a fairer comparison?```
> > >
> > > Thanks for bringing up potential modifications for the simple ensemble baseline. We agree that an ensemble size of 3 looks small. We are running experiments with ensemble size of 5 and 10 for the ConnectFour board game scenarios. In fact, these two experiments will take several days to finish for one run due to the limit on computation resource we currently have. From the preliminary results (first few million steps), we do not see a substantial improvement over the 3 policy ensemble. But we do acknowledge there could be a substantial improvement given a much larger size ensemble (order of hundred) as suggested in Czarnecki, 2020. Our new experiment results with ensemble sizes 5 and 10, and our new IET results, (see our response to reviewer obGm for more detail)  show that standard ensemble of size 10 may have the potential of achieving similar performance as IET. However, within the limited number of training time we have run (about 148hrs), the best testing reward achieved by the standard ensemble baseline with 10 policies (-0.11) is still far below the best testing reward achieved by our IET approach (0.99, >0.9 after 53hrs' training). As for the suggestion on using Nash distribution over the policies, we would like to request the reviewer to elaborate on how specifically this should be done. We ask this because suppose we are to use Nash distribution in training (by picking only the policies within the Nash distribution to train against each other), it is likely that once one policy becomes dominant within the ensemble due to randomness, then only this policy is trained and it becomes even more dominant (note that at early stage of training, policies may not be very strong, so policy that is trained more is likely to dominate). This type of instability may cause the ensemble to collapse into a single policy. PSRO finds the best response to the Nash distribution as the new policy to be added to the existing set of (frozen) policies, so it strictly increases the diversity of the ensemble due to the growing size. But with a fixed size ensemble, this argument may not apply. Another possibility is to use Nash distribution for testing. But since the Nash distribution is solved with respect to the opponent ensemble in the training, we suspect this probably does not result in substantially better result against the exploiter opponent in the testing since the exploiter is out of the training distribution, so even the Nash distribution is not guaranteed to the most robust mixed strategy against an out-of-distribution opponent (not sure if this statement is correct, please correct if we are wrong).
> > >
> > > ```● Personally, I found figures 2,3,4 confusing to understand most notably due to their poor legends.```
> > >
> > > Thanks for the feedback, we are working on updating these figures, also based on the feedback from the other reviewers.

---

### Review · Reviewer_obGm · 2022-08-04

**Summary Of Contributions:**

This paper addresses the problem of robustness to distribution mismatches between training and testing in multi-agent settings. Previous works demonstrated the effectiveness of ensemble training to overcome the distribution mismatch, but the computation and memory complexity scale linearly with the size of the ensemble. This paper proposes an implicit ensemble training approach, called IET, in which a single policy architecture is conditioned on a latent variable to produce diverse behaviors, effectively simulating an ensemble of policies. This simple procedure is evaluated on a series of board games and continuous control tasks, in which IET shows robustness to distribution mismatches and improved sample efficiency thanks to parameters sharing.

**Broader Impact Concerns:**

I do not believe this paper requires to explicitly address the potential ethical implications of the work.

**Requested Changes:**

Critical

- (Significance of the Results) I encourage the authors to detail the number of seeds employed in the experiments, and to comment the statistical significance of the reported results.

Somewhat Critical

- (Main Experimental Result) To my understanding, the main message the paper is trying to convey is that IET could provide similar robustness w.r.t. distribution mismatches as training with large ensembles, while avoiding the computation and memory blow-up. However, the experimental analysis only provides a comparison with a rather small ensemble (of three policies). I would suggest the authors to center their experimental section on a main experiment that compares the robustness achieved by IET with standard ensembles of increasing size. It would be especially crucial to show how many policies are needed to match the robustness of IET, while detailing the computation and memory requirement of the explicit ensemble.

Strengthening the Work

- (Analysis of the End-to-End Training) I am not completely convinced it is safe to train the diverse module end-to-end. What is preventing the diverse module to collapse to a Dirac delta, especially if the game admits a pure strategy equilibrium? I think that a deeper discussion on the pros/cons of end-to-end training could nicely complement the presentation of the IET approach.

- (Minmax Comparison) I was actually expecting the comparison against the MT baseline to act as a sanity check, i.e., to show that IET is not far off the robustness that can be achieved by an "ideal" method, while being much more efficient in practice. Instead, the comparison against MT is quite unclear, and it seems to underline the flaws of the baseline rather than the merits of IET. I would suggest the author to provide a comparison in a much simpler domain where MT can perform well, or to move the comparison to the appendix to free some space for more interesting results.

**Strengths And Weaknesses:**

Strengths
- (Simplicity of the Approach) The paper proposes a simple and reasonable approach to achieve robustness to distribution mismatches while preserving sample and computation efficiency;
- (Experimental Analysis) The experimental analysis includes various domains and a rich set of metrics, illustrations, and ablations;
- (Clarity of the Presentation) The paper is well-written and easy to follow;
- (Related Work) The paper does a good job in relating the presented contributions with previous works.

Weaknesses
- (Significance of the Experiments) The statistical significance of the reported results is mostly unclear. The text does not explicitly state the number of seeds employed in the experiments, as well as the meaning of shaded areas in the plots, performance ranges in the tables;
- (Clarity of the Results) The experimental results are not always strong. The experimental section tries to convey too many information instead of focusing on a few crucial results.

General Comment

The paper tackles a relevant problem with a very reasonable approach, by borrowing ideas from multi-task learning to improve the efficiency of ensemble training. Having read the first three sections I was sold on the value of the methodology and the benefits it can provide. However, the experimental section is slightly underwhelming, as it struggles to clearly validate the promises of the previous sections, perhaps trying to convey too many information. While I am not an expert in the multi-agent domain, I think this paper provides a valuable contribution overall, even if the paper could be improved in a couple of aspects (more details below).

Minor Comments
- In Figure 2, it is not immediately clear what is changing between the top row and the bottom row of plots;
- There is a duplicated reference for (Czarnecki, 2020).

---

> ### Author Response · Authors · 2022-08-20
> **Responses**
>
> `(Significance of the Results) I encourage the authors to detail the number of seeds employed in the experiments, and to comment the statistical significance of the reported results.`
>
> Thanks for the suggestion. We will add the following detail about the experiment results to elaborate on the statistical significance. Specifically, in the original manuscript, we ran each training setting with 3 random seeds. However, we did not average the reward curves over the random seeds, but instead, only selected the one with the highest testing reward at the end of the training, along with the corresponding training reward, for the plot. We made this choice because we found that the reward curves across different random seeds may exhibit quite distinct (somehow random) temporal pattern (w.r.t. the number of training steps). We suspect that this observation is a result of the difficulty to balance the relative strength between the two agents in adversarial games. We think that the absence of a consistent pattern across random seeds suggests that average across random seeds may not be a sensible choice for showing the evolution of training and testing rewards w.r.t. the training steps. The shaded areas in the plots denote the standard error of the temporally smoothed reward curves (since we using stochastic policies, the rewards are also random variables; besides, Leduc and Texas Hold'em also have intrinsic randomness due to chance nodes).
>
> `(Main Experimental Result) To my understanding, the main message the paper is trying to convey is that IET could provide similar robustness w.r.t. distribution mismatches as training with large ensembles, while avoiding the computation and memory blow-up. However, the experimental analysis only provides a comparison with a rather small ensemble (of three policies). I would suggest the authors to center their experimental section on a main experiment that compares the robustness achieved by IET with standard ensembles of increasing size. It would be especially crucial to show how many policies are needed to match the robustness of IET, while detailing the computation and memory requirement of the explicit ensemble.`
>
> Thanks for the suggestion, we include here additional experiment results with the standard ensemble baseline with 5 (1 random seed) and 10 policies (4 random seeds) for the Connect Four scenario. Here is a summary of the results (reported training time is all with 6 cores on a 48-core machine with two AMD Ryzen Threadripper 3960X 24-Core Processors, with 128GB ram): For standard ensemble with size 5, we ran 134M steps, which took 138hrs (1.03 hr/Mstep), the final [testing agent reward](https://drive.google.com/file/d/14ymfhEegFjmtvf1HNlQhlR898-j4EAZw/view?usp=sharing) is -0.79; For standard ensemble with size 10, we ran 87M steps, which took 148hrs (1.70 hr/Mstep), the final [testing agent rewards](https://drive.google.com/file/d/1ZmiHqKkN7n5siCFZlqb4uhK1mtxfPk1B/view?usp=sharing) are (ranked from high to low) [-0.12, -0.18, -0.32, -0.44], with an average testing reward -0.27. For our IET approach, we have run a total of 6 additional random seeds. We ran 60M steps for 4 random seeds, which took 68hrs (1.13 hr/Mstep); and 120M steps for 2 random seeds, which took 140hrs (1.17 hr/Mstep). At 60M steps, the testing rewards are [0.97, -0.44, -0.47, -0.61, -0.65, -0.88], with an average reward -0.33. At 120M steps, the testing rewards are
> [0.99, 0.95], with an average testing reward 0.97. The two random seeds that have been run for a total of 120M steps reached a testing reward >=0.9 and stayed above that after 45M and 99M steps, respectively. From the reward curves, it looks like the standard ensemble with 10 policies runs and our IET runs that were terminated at 60M steps might not have converged to a steady state yet. From all the experiments that we have ever run with the standard ensemble baseline up to 10 policies, and up to 134M steps, we have not seen any that has achieved a stable testing reward above 0.0 (indicating the agent always loses to the exploiter adversary on average). In contrast, our IET approach is able to achieve near 100\% winning rate against the exploiter adversary in 2 out of 6 runs.
>
> Based on the discussion above, we would like to improve the experiment settings in the following aspects: (1) Larger total training steps: the performance of IET does not seem to have converged within 50M training steps in the original experiment setting; (2) Standard ensemble with more policies: we will add results for standard ensemble with size 5 and size 10; (3) Statistic analysis with more random seeds: we will increase the number of random seeds from 3 to 10 for each training settings.

---

> > ### Author Response · Authors · 2022-08-20
> > **Responses continued**
> >
> > `(Analysis of the End-to-End Training) I am not completely convinced it is safe to train the diverse module end-to-end. What is preventing the diverse module to collapse to a Dirac delta, especially if the game admits a pure strategy equilibrium? I think that a deeper discussion on the pros/cons of end-to-end training could nicely complement the presentation of the IET approach.`
> >
> > Thanks for the comment and suggestion. We agree with the speculation that the conditioning variable distribution may collapse into a Dirac delta distribution if the game is fully transitive (a.k.a. Elo games, see Czarnecki, 2020), where training with the single policy setting should also converge to the strongest policy. In this case, it is desired for the output distribution of the shaping network to collapse into a Dirac delta distribution. In more complicated games where there exist a large set of highly intransitive skills, however, mode collapse is undesireable. As in the response to Reviewer mj1G, we argue that the simple objective of maximization of agent's own cumulative reward subjective to the joint state transition induced by the joint policy could naturally drive the shaping network to output a non-collapsing distribution (this argument is also supported by our new experiment results above and the gradient analysis in our response to Reviewer Cc15, although this is not always guaranteed due to the intrinsic difficulty of adversarial training as discussed). We have also considered explicitly adding auxiliary loss to avoid mode collapse as suggested by Reviewer Cc15, but the trade-off is potential sub-optimality when the game is highly transitive. One significant result of our work is that we have demonstrated that without any explicit signal for manipulating policy diversity, our IET network is able to learn an implicit ensemble that has the required level of diversity to achieve good generalization performance for the game it is trained on. We will add a paragraph at the end of the Approach section to discuss the pros/cons of our design choice.
> >
> > `(Minmax Comparison) I was actually expecting the comparison against the MT baseline to act as a sanity check, i.e., to show that IET is not far off the robustness that can be achieved by an "ideal" method, while being much more efficient in practice. Instead, the comparison against MT is quite unclear, and it seems to underline the flaws of the baseline rather than the merits of IET. I would suggest the author to provide a comparison in a much simpler domain where MT can perform well, or to move the comparison to the appendix to free some space for more interesting results.`
> >
> > Thanks for the suggestion. We have also tried hard to find a domain where the MT may be a strong baseline for comparison, but failed. In the original MT paper, the MT approach achieves marginally higher performance against the MADDPG baseline in the multiagent particle environment. Our conjecture is that although MT relies on a theoretically sound minimax formulation, the introduced one-step gradient approximation might be too crude to achieve the theoretical advantage of minimax training. We think an additional baseline using NeuPL as suggested by Reviewer mj1G would be a better comparison. We would like to add this experiment comparison once we have it.

---

> > > ### Comment · Reviewer_obGm · 2022-08-22
> > > **Re: Responses**
> > >
> > > I want to thank the authors for their thorough replies, which clarified some of my previous doubts.
> > >
> > > My major remaining concern is related to the statistical significance of the reported results. I am not sure taking the best performing curve across different seeds is a sound metric, while the odd temporal pattern of the learning curves might suggest it is better to confront the performance at convergence rather than the curves themselves. Moreover, the point made by Reviewer Cc15 over the differences in the architectures of IET and the baselines seems critical.
> > >
> > > Thus, I would ask the authors to provide additional experimental results comparing average performances at convergence across a significant number of seeds, and to possibly include these new results directly in the manuscript.

---

> > > > ### Author Response · Authors · 2022-09-02
> > > > **Responses**
> > > >
> > > > We would like to thank the reviewer again for the thoughtful feedback! We have included new results in the updated manuscript in response to the reviewer's concern (statistical significance). Please kindly check out the updated pdf as well as [our new response to Reviewer Cc15](https://openreview.net/forum?id=LfTukxzxTj&noteId=YgI4NsjmUF) which should already include the detail to address the raised concern.

---

> > > > > ### Comment · Reviewer_obGm · 2022-09-02
> > > > > **Recommendation**
> > > > >
> > > > > I want to thank again the authors for addressing my concerns on the experimental analysis and for updating the paper.
> > > > >
> > > > > Although I see a remarkable improvement in the presented results, I am still skeptical on their statistical significance. On the one hand, I think that the practice of filtering out "bad seeds" should be avoided, as it might hamper the fairness of the comparison. On the other hand, I do not understand how the best performance (bolded) has been selected in Table 1. The confidence intervals (how they are computed?) overlap, which makes the performance difference not significant. Considering more seeds might help obtaining a clearer comparison.
> > > > >
> > > > > The statistical significance of the results presented in deep RL papers has been debated recently (e.g., Henderson et al., Deep reinforcement learning that matters, 2018; Agarwal et al., Deep reinforcement learning at the edge of the statistical precipice, 2021), and I believe it cannot be overlooked unfortunately. For this reason, I am providing a slightly negative evaluation for the work as is. However, I will make clear to the editor that I believe this will be a valuable paper with a more solid experimental analysis.

---

> > > > > > ### Author Response · Authors · 2022-09-03
> > > > > > **Response**
> > > > > >
> > > > > > Thanks for the thoughtful comments and concern! Here are our response which we hope will help the reviewers and readers better understand the presented results.
> > > > > >
> > > > > > ```
> > > > > > On the one hand, I think that the practice of filtering out "bad seeds" should be avoided, as it might hamper the fairness of the comparison.
> > > > > > ```
> > > > > > We made this choice based on the following reasoning: The intent is to evaluate the generalization performance of the agent against a previously unseen exploiter. To achieve this, we look at the testing reward of the agent against the exploiter. However, a low testing reward may not necessarily suggest that the generalization performance is poor, because there could be two different situations that lead to a poor testing reward: (1) The training reward is high, but the agent overfits to the training adversary, which leads to a low testing reward against the exploiter; (2) The training reward is low (possibly due to imbalance in adversarial training), and therefore we cannot anticipate a testing reward that is higher than this low training reward. The argument is that the testing reward is indicative of the generalization performance only in the first situation, but not in the second situation, so we filter out those results that have low training reward.
> > > > > >
> > > > > > That being said, we agree that it might be a little unfair in the sense that the capability of achieving balanced training with a high probability is also an important metric for the approach to be useful in practice. Therefore, we think it would be beneficial to explore approaches to improve the training stability of our IET approach to further improve the usefulness. We briefly discuss a few thoughts: (1) One possible approach could be designing a feedback controller for the training loop that assigns unequal numbers of alternating training steps to the two agents with the control objective of balancing the training reward. This approach is orthogonal to the ensemble training approaches. (2) Another approach could be designing an entropy or mutual information regularizer (as suggested by Reviewer Cc15) to the learned distribution of the latent condition variable at early stage of training and gradually anneal this regularizer's weight to zero in order to reduce the probability of mode collapsing at early stage of training while not preventing the convergence to a strong pure strategy at late stage of the training, with the hope that avoiding mode collapsing can help balance training. We would like to explore these thoughts in future works.
> > > > > >
> > > > > > ```
> > > > > > On the other hand, I do not understand how the best performance (bolded) has been selected in Table 1. The confidence intervals (how they are computed?) overlap, which makes the performance difference not significant. Considering more seeds might help obtaining a clearer comparison.
> > > > > > ```
> > > > > > The best performances were selected based on the mean only. We agree that this might not be very reasonable if the confidence intervals overlap. Therefore, we decided to update the manuscript by unbolding the metrics where the intervals overlap. The confidence intervals are calculated as the standard error of mean across valid seeds. We agree that it would be helpful to run with more seeds such that the confidence intervals no longer overlap to improve the confidence of the results.
> > > > > >
> > > > > > ```
> > > > > > The statistical significance of the results presented in deep RL papers has been debated recently (e.g., Henderson et al., Deep reinforcement learning that matters, 2018; Agarwal et al., Deep reinforcement learning at the edge of the statistical precipice, 2021), and I believe it cannot be overlooked unfortunately. For this reason, I am providing a slightly negative evaluation for the work as is. However, I will make clear to the editor that I believe this will be a valuable paper with a more solid experimental analysis.
> > > > > > ```
> > > > > >
> > > > > > Yes, we agree that statistical significance is very important for a faithful evaluation of deep RL approaches, where the randomness is much more prominent than in the supervised learning domains. We will try to improve the statistical significance of the results with more experiment runs. We would like to thank the reviewer again for all the thoughtful comments and feedback, which helped us a lot to improve the quality of our submission.

---

### Review · Reviewer_Cc15 · 2022-08-06

**Summary Of Contributions:**

The authors introduce a more compute-efficient way to induce policy diversity in agents and show that these agents show increased robustness in the face of novel players in multi-agents games (MARL). Specifically, they introduce implicit ensemble training (IET), which parameterize an ensemble as a goal-conditioned network with the goal coming from a source of external noise. They compare this approach to single policy agents, simple ensembles (i.e. N discrete networks), and adversarial trained agents on several two player adversarial games. They also analyze the effect of cross-player parameter sharing, though how this connects to their primary contribution is somewhat unclear.

**Requested Changes:**

level of criticality: ** -> **** (least to most critically important for acceptance)

**** Evidence that IET helps test-time performance when controlling for train-time performance

*** Evidence that IET yields a diverse set of policies, at least comparable to explicit ensemble methods.

** Replace 4.6.2. with ablations of IET

** Remove the T-SNE stuff or explain how your interpretation is justified

** Improved experimental presentation as per 3)

**Strengths And Weaknesses:**

## Strengths
* The introduction to MARL was well-written
* The problem statement was quite clear and well-motivated -- I agree that we need more compute-efficient ensembles
* The approach chosen seems simple to implement
## Weaknesses

1) Everything in your problem framing suggests that IET shouldn't help training time performance: a single policy as as expressive for best-response. IET should *only* help for notions of robustness and generalization. However, your results show that IET does significantly boost performance during training (perhaps more so than at test time...). This can be explained by the large differences between the network architecture of IET and the baseline methods. Ideally, all methods should use the same architecture. Since the modular architecture used by IET is inherently multi-task, perhaps keeping the conditioning variable constant for baseline methods would be sufficient.

2) Prior work has shown the general difficulty in learning diverse sets of policies, even in the single-agent regime (e.g. "Diversity is all you need"). I'm highly credulous that merely conditioning on a random variable will yield policies that are diverse wrt that random without any explicit objective pushing it to do so (e.g. the mutual information between the random variable and the resultant behavior). The only evidence you show for policy diversity is the increased performance (which could have other causes e.g. 1) ) and the T-SNE plots of the random variable in different games. Interpreting the complexity of T-SNE images is already a fraught undertaking, but even if we assume the conditioning variable is more diverse in harder environments, that isn't what we care about. We care about policy diversity: the extent to which a change in the conditioning yields a meaningful change in behavior. Without such evidence, how can we be sure that your explanation of the performance improvements is correct?

3) The experimental results could be better formatted. The top and bottom rows of Figures 2-4 should be redundant, since one player's performance fully determines the performance of the other player, so the bottom rows should be removed. I'm a bit confused why they aren't perfectly symmetric e.g shouldn't Figure 3a bottom just be the negation of Figure 3a top? The axes aren't shared across columns, making cross-method comparison difficult -- why not just have a single figure comparing IET to both baselines?

4) How does the 4.6.2 tie into the rest of the paper? Cross player parameter sharing is another way to reduce computation, but it seems orthogonal to IET and I'd suggest removing it.

5) Ablate aspects of your approach. Specifically, IET without modular networks and IET without the shaping network. Choices for the input noise (why Gaussian, not categorical?) and shaping network output activation should at least be explained and potentially ablated.

---

> ### Author Response · Authors · 2022-08-15
> **Responses**
>
> `Everything in your problem framing suggests that IET shouldn't help training time performance: a single policy as as expressive for best-response. IET should only help for notions of robustness and generalization. However, your results show that IET does significantly boost performance during training (perhaps more so than at test time...). This can be explained by the large differences between the network architecture of IET and the baseline methods. Ideally, all methods should use the same architecture. Since the modular architecture used by IET is inherently multi-task, perhaps keeping the conditioning variable constant for baseline methods would be sufficient.`
>
> We agree with the insightful statement that the difference in the training performance can be due to the different network architecture, and it would be helpful to include a baseline with a constant conditioning variable for justifying the benefits of randomizing the condition variable. We indeed did that and the results showed that the multitasking network with constant conditioning variable exhibit similar behavior as the single policy training baseline: poor testing performance. However, we also observed that the modular multitasking settings (with gaussian latent noise or with fixed latent noise) in general have larger reward fluctuation along the axis of training steps compared to the simple ensemble settings (single policy training included, as a special simple ensemble with ensemble size 1) with feedforward network. We suspect that this is because the modular network is deeper and has more complicated structure than the feedforward network, and thus more difficult to optimize, which introduces additional randomness (that has a longer time scale than the randomness caused by action sampling) to the training. As for the high training reward of our IET settings shown in Fig. 2,3,4, we provide the following facts and analysis for a better understanding of this result: (1) For the modular multitasking settings (with gaussian latent noise or with fixed latent noise), we observed that the training reward does vary depending on random seeds. (2) In the Connect Four (CF) scenario, for some random seeds, the training reward may go up to near 1.0 (where the testing reward may range from 1.0 to -1.0), or go down to near -1.0 (where the testing reward also goes down to near -1.0). We conjecture that this sensitivity could be explained by the difficulty of optimizing this network as well as the general difficulty of balancing the relative strength between agents in adversarial games. We will provide further analysis and evidence by looking into the gradient of the output action logits w.r.t. the conditioning variable (we will elaborate on this in the response to the next comment) to support our conjecture. (3) The CF scenario is asymmetric for the first player (agent) and the second player (adversary), where the agent has advantage (won't lose if playing optimally).
>
> `Prior work has shown the general difficulty in learning diverse sets of policies, even in the single-agent regime (e.g. "Diversity is all you need"). I'm highly credulous that merely conditioning on a random variable will yield policies that are diverse wrt that random without any explicit objective pushing it to do so (e.g. the mutual information between the random variable and the resultant behavior). The only evidence you show for policy diversity is the increased performance (which could have other causes e.g. 1) ) and the T-SNE plots of the random variable in different games. Interpreting the complexity of T-SNE images is already a fraught undertaking, but even if we assume the conditioning variable is more diverse in harder environments, that isn't what we care about. We care about policy diversity: the extent to which a change in the conditioning yields a meaningful change in behavior. Without such evidence, how can we be sure that your explanation of the performance improvements is correct?`
>
> We would like to thank the review for this critical comment. Inspired by this comment, we did further analysis on the relationship between the latent noise and the output policy, which greatly deepened our understanding of the capability and limitation of our IET approach, as well as the results.
>
> We acknowledge the difficulty in learning diverse set of policies in single-agent regime. However, we argue that it is not unreasonable to anticipate diversity to automatically emerge given enough model capacity. In fact, multiagent scenarios with high intransitivity of skills should naturally incentivize agents to learn diverse behaviors because the `optimal behavior' is a mixed strategy equilibrium, while in single-agent regime the optimal policy is a single deterministic policy. Previous works, such as (Bansal, 2017; Czarnecki, 2020) have also demonstrated learning of emergent (strong) behaviors simply by optimizing the cumulative reward using an ensemble with sufficient number of policies.

---

> > ### Author Response · Authors · 2022-08-15
> > **Responses Continued**
> >
> > To confirm that our IET is able to learn a diverse set of policies, we did additional analysis which provides further insights on the capability and limitation of our current formulation. We evaluated the gradient of the action logits w.r.t to the latent noise ($G = \sum_{i,j}  |\frac{\partial p_i}{\partial w_j}|$, where $p_i$ is the action logits, and $w_j$ is the component of the latent noise vector). $G$ measures how much the policy is influenced by the latent noise vector. To eliminate the other randomness, we evaluated $G$ on the Connect Four scenario since this game has deterministic initial state and state transition. We show the histogram of $G$ evaluated using the state corresponding to the first time step of Connect Four (always empty board) across 500 runs. We identified three different patterns of the histogram (across different runs of our IET approach with Gaussian latent noise or with fixed latent noise), each of which corresponds to a certain type of training and testing reward pattern: (1) The [first type of histogram pattern](https://drive.google.com/file/d/1zY9Cm2sPJLoxv6kOvAPfuKDK7Nb1-iTV/view?usp=sharing) corresponding to mode collapse, where $G$ is always 0. This happens for a few runs with Gaussian noise and all the runs with fixed latent noise. Under this situations, the training and testing rewards have similar pattern as in the single policy training setting (with a testing reward within a rough range from -1.0 to -0.5, and a training reward varying within a rough range -0.5 to 0.5), [example, between 1.5e7 to 5e7 steps](https://drive.google.com/file/d/1oBrG1GmmJpOsDom7kJkrDTqg_WsFxUbY/view?usp=sharing); (2) The [second type of histogram pattern](https://drive.google.com/file/d/1Gi5pn0YI9kkBcMxeurHM5zS1hb9n5W8k/view?usp=sharing) has the majority of probability mass at $G=0$, as well as quite a few probability masses at a variety of discrete values of $G$ ranging from 0 to $\sim$ 15 (this range also varies with different random seed, but roughly on the same scale). The interpretation of this pattern is that the agent learned to play a mixed strategy by adopting a certain policy most of the time ($G=0$, latent noise is neglected), but occasionally playing other different policies ($G \neq 0$ action probability is dependent on latent noise). We found that the testing reward under this situation is always reasonably good (range from -0.4 to 1.0), and the training reward is always higher than the testing reward,  [example1](https://drive.google.com/file/d/1ld0ukJHNHmI3N2rg5rAS0SzX1Obr539t/view?usp=sharing) and [example2](https://drive.google.com/file/d/1r3K_FToTtA2TAdHG7_D3vlAp9fl0l1rE/view?usp=sharing); (3) The [third type of histogram pattern](https://drive.google.com/file/d/1WjiQ8lfg5aOn3s5ZHGTaTNElN91eeesy/view?usp=sharing) has a much denser support and long tail than the second type with $G$ ranging from 0 to $\sim$ 500 (roughly). The large scale of $G$ suggests that the action logits are very sensitive to the latent noise (overfitting to the noise). Both the training reward and testing reward are close to -1 under this situation, [example, between 4e7 to 6e7 steps](https://drive.google.com/file/d/13HxS8TKBLMFfV3zk1PvYvtiNBhaSDvSt/view?usp=sharing).
> >
> > To summarize, from the gradient analysis, we verified that our IET is able to learn a mixed strategy conditioned on the latent noise, which has good generalized performance against an exploiter agent, without explicit loss term to encourage this conditioning. However, this learning of conditional policy is not always successful due to potential difficulties such as mode collapse and overfitting to the latent noise, which could be related to the general difficulty of training imbalance in adversarial settings. Therefore, one future direction could be investigating effective method to balance the training (we have not yet done an exhaustive hyper-parameter tuning, we believe selecting better choices for a few hyper-parameters such as reducing the number of consecutive episodes between alternatingly training the two agents could help balance the training, which we have not tried yet given limited computation resource.) or directly encourage the policy to condition on the latent noise (we will also discuss the downside of explicitly doing so in the response to Reviewer obGm's comment) so as to improve the success rate of learning strong policy.
> >
> > Based on the above discussion, we will make the following modifications to our manuscript: (1) We will add the baseline result of IET with fixed latent noise for comparison; (2) We will add the gradient analysis results with a discussion on the potential failure cases of our IET approach; (3) We will update the T-SNE graph with color intensity denoting the magnitude of $G$, which should provide additional information on how the policy conditions on different partitions of the latent noise; (4) We will add a paragraph discussing future directions on improving the training of our IET approach.

---

> > > ### Author Response · Authors · 2022-08-16
> > > **Responses continued**
> > >
> > > `The experimental results could be better formatted. The top and bottom rows of Figures 2-4 should be redundant, since one player's performance fully determines the performance of the other player, so the bottom rows should be removed. I'm a bit confused why they aren't perfectly symmetric e.g shouldn't Figure 3a bottom just be the negation of Figure 3a top? The axes aren't shared across columns, making cross-method comparison difficult -- why not just have a single figure comparing IET to both baselines?`
> > >
> > > Thanks for the suggestion on the format, we agree with the comments, and will update the plots accordingly. The agent reward and the adversary reward are not perfectly symmetric because of an artifact in the implementation of the policy: we did not explicitly mask out illegal actions. The default setting in RLcard is that when an agent takes an illegal action, this agent is penalized by getting a negative reward of -1. As a result, the sum of agent and adversary rewards (absolute value) equal to the average frequency of an illegal action being taken. This value gradually increases to near 0 as the training goes on, during which both agents learn to avoid taking illegal actions. But we agree that the marginal information of adding the adversary reward is very low given that we have already shown the agent reward, so we will remove the adversary reward in the update.
> > >
> > > `
> > > How does the 4.6.2 tie into the rest of the paper? Cross player parameter sharing is another way to reduce computation, but it seems orthogonal to IET and I'd suggest removing it.`
> > >
> > > Thanks for the feedback. Section 4.6.2 shows a spectrum of level of sharing and the trade-off between computational efficiency and robustness. Table 7 illustrates the representation power and parameter efficiency as compared to alternative ensemble approaches with parameter sharing. We believe this discussion is beneficial, which can help the readers to get a better understanding of our IET approach and how the multitasking network plays an important role in our approach. We also received a positive feedback from Reviewer mj1G, which consolidated our belief on that point. We will think about rephrasing this section to make it look more closely related to the rest of this paper.
> > >
> > > `Ablate aspects of your approach. Specifically, IET without modular networks and IET without the shaping network. Choices for the input noise (why Gaussian, not categorical?) and shaping network output activation should at least be explained and potentially ablated.`
> > >
> > > Thanks for the suggestions on the various aspects of ablation. We will add results of IET with the modular network replaced by a MLP, and IET without the shaping network, as suggested. We choose Gaussian noise instead of categorical random variable because Gaussian noise is continuous and thus can take a continuum of values. This property suggests that our IET approach can potentially represent a mixed strategy with unlimited number of pure strategies, while the actually required complexity of the mixed strategy is learned end to end (adaptive to the transitivity nature of the game). In contrast, categorical random variable can only take a finite number (another hyperparameter to be tuned depending on the transitivity nature of the game) of values, and therefore representing a mixed strategy with a finite number of pure strategies.
> > > One justification for the choice of using a softmax activation for the output of the shaping network is from this mixed strategy interpretation (the shaping network is responsible for learning some embedding that corresponds to the mixture coefficients, while the modular network is responsible for learning the pure strategies). In the original soft-modular network paper (Yang et al., 2020), the embedding input is a one-hot encoding of the task id. The output of the softmax activation could be interpreted as a continuous embedding corresponding to a mixture of tasks. However, we agree that this interpretation does not necessarily lead to this softmax design choice. We will add an ablation study by removing this activation and/or replacing it with other type of activation functions to better understand its role.

---

> > > > ### Comment · Reviewer_Cc15 · 2022-08-22
> > > > **Paper Revision and/or more details?**
> > > >
> > > > Thank you for your detailed response. I particularly enjoyed your response to my concern over diversity emerging without additional loss terms. I see now that I was fixated on the single-agent setting -- I agree that such emergence is possible in the multi-agent settings (which is what this paper is concerned with) and retract my previous criticism.
> > > >
> > > > However, it's quite hard to judge the merits of many of your suggested changes without them being incorporating into the paper (or at least providing more details and quantitative results). For example, your response to "Everything in your problem framing suggests that IET shouldn't help training time performance..." was basically a description of new experimental results. Showing me the actual results would be much more convincing.
> > > >
> > > > Your new experiments attempting to show policy diversity are interesting, but I don't think your chosen metric is particularly meaningful and it doesn't seem like you've got an objective method for sorting these histograms into categories (showing the histograms for seeds in each test-performance quartile would be one such method). Your measure shows some effect on the policy, but not necessarily a meaningful effect e.g. if the latent noise value changed p(act=1 | state) = 0.001 to p(act=1 | state) = 0.0001, that would still count towards your measure without significantly changing the policy. More generally, there could be local correlations between the policy and the latent noise that don't manifest into meaningful changes across a full game e.g. latent noise value decides between 2 actions with identical effects on the environment.
> > > >
> > > > Here's a simpler metric that I think would be a lot easier to interpret: at test time, compare the performance of the full IET agent (i.e. resampling there latent noise vector each game) vs the same IET agent with its latent noise vector held constant throughout  this evaluation. Under your hypothesis, the resampling agent should yield significantly higher performance.

---

> > > > > ### Author Response · Authors · 2022-09-02
> > > > > **Responses**
> > > > >
> > > > > ```
> > > > > However, it's quite hard to judge the merits of many of your suggested changes without them being incorporating into the paper (or at least providing more details and quantitative results). For example, your response to "Everything in your problem framing suggests that IET shouldn't help training time performance..." was basically a description of new experimental results. Showing me the actual results would be much more convincing.
> > > > >
> > > > > Your new experiments attempting to show policy diversity are interesting, but I don't think your chosen metric is particularly meaningful and it doesn't seem like you've got an objective method for sorting these histograms into categories (showing the histograms for seeds in each test-performance quartile would be one such method). Your measure shows some effect on the policy, but not necessarily a meaningful effect e.g. if the latent noise value changed p(act=1 | state) = 0.001 to p(act=1 | state) = 0.0001, that would still count towards your measure without significantly changing the policy. More generally, there could be local correlations between the policy and the latent noise that don't manifest into meaningful changes across a full game e.g. latent noise value decides between 2 actions with identical effects on the environment.
> > > > > ```
> > > > >
> > > > > Thanks for the insightful feedback! We updated the manuscript with a few new key results in as response (please kindly refer to the updated/new Table, Fig. 2, Fig. 3, Section 4.4.1, and 4.6.2, and corresponding paragraphs highlighted by blue text) for detail. We briefly summarize the new experiments/findings: (1) We ran more experiments to improve the statistical significance of the results (see updated section 4.4.1, and Table 1), and compare our IET with latent-noise-fixed IET and standard ensemble of size 10. The new results show that our IET outperforms both baselines, especially for the best testing reward across random seeds. (2) We visualize the histogram of action probabilities of our IET agent and the standard ensemble agent of size 10, at various snapshots across the training (see updated section 4.6.2, Fig. 2 and 3). The results suggest that: (i) Our IET approach does represent a mixture of pure strategies (Fig. 3c and 3d); (ii) IET agent does not always learn mixed strategy (Fig. 3a, 3b, 3e, 3f), which could correspond to mode-collapsing (very weak policy, Fig. 3a, 3b) or converging to a strong (near optimal) pure strategy in fully observable domain (Fig. 3e, 3f). Our finding is in accordance with the findings in Czarnecki et al. on the geometric structure of the Connect Four game. (iii) IET agent could represent much more diverse policies than the standard ensemble agent does (Fig. 3g, 3h). As a result, our IET achieves higher testing reward.
> > > > >
> > > > > ```
> > > > > Here's a simpler metric that I think would be a lot easier to interpret: at test time, compare the performance of the full IET agent (i.e. resampling there latent noise vector each game) vs the same IET agent with its latent noise vector held constant throughout this evaluation. Under your hypothesis, the resampling agent should yield significantly higher performance.
> > > > > ```
> > > > >
> > > > > Thanks for this nice suggestion! We appreciate it a lot, and we agree that this sounds very sensible. However, after trying it out and getting more insight from the results, we found that this statement is actually not true. The fact is that diversity is necessary for learning strong policy, but once IET finds strong policy, the resulted strong policy is not necessarily diverse (in fact, strong policy tends to lack diversity). This statement is also justified by the findings in Czarnecki et al. (Please refer to the updated Section 4.4.1 in our manuscript for detailed discussion).
> > > > >
> > > > > Due to the computation resource constraint, we will keep working on addressing the other requested changes.

---

### Decision · Action_Editors · 2022-09-30

**Recommendation:** Accept with minor revision

**Comment:**

The paper studies how to address distribution robustness for multi-agent RL (MARL) setting against novel opponents, and proposes to replace network ensembles, which could be computationally expensive to scale, with a single latent conditioned policy with external noise.

Two reviewers recommended "leaning accept" while one reviewer recommended "leaning reject". A "leaning accept" also expresses the decision to be borderline. Given the recommendations, my proposal is "Accept", mainly for:

- this is a simple easy-to-understand/apply method with clear audience (MARL). other papers may take this method as future baselines.
- reviewers appear satisfied with the efforts in the rebuttal: e.g. Reviewer Cc15's many feedback were addressed with new experiments (e.g. Jacobian norm), Reviewer mj1G's feedback on lack of larger ensembles as baselines was addressed with new experiment with 5/10 ensembles and additional related work were discussed, and Reviewer obGm's concern on statistical significance is partially addressed.

However, this "Accept" is borderline, and there are worries about the work. Please provide these two additional revisions:
- statistical significance of the results: the results are quite noisy and the whole validity of method/domain seems to be questioned by Reviewer obGm recommending "leaning reject" as well as Reviewer Cc15 recommending "leaning accept" with a careful note. Could you follow up with additional experiments to engage in convincing these two more about statistical significance of the results? e.g. even running on more seeds or qualify the claims.
- lack of discussion on literatures on efficient ensemble: Replacing ensembles with a stochastic network is not novel. It has been debated/studied in many papers on Bayesian deep learning (sometimes applied to sparse-reward exploration problems in RL), e.g. dropout [1] vs bootstrapped DQN [2]. These papers and other work should be referenced and added to the related work section.

[1] Gal, Yarin, and Zoubin Ghahramani. "Dropout as a bayesian approximation: Representing model uncertainty in deep learning." international conference on machine learning. PMLR, 2016.

[2] Osband, Ian, et al. "Deep exploration via bootstrapped DQN." Advances in neural information processing systems 29 (2016).

---

> ### Author Response · Authors · 2022-10-04
> **Thanks to the reviewers and the AE**
>
> We would like to thank the reviewers again for all the insightful feedback and discussions, which not only greatly helped us improve the quality of the manuscript, but also deepened our understanding of the pros and cons of the proposed approach. Thanks to the discussion, we have also had some pretty good ideas of a few promising future directions to extend our current approach.
>
> We would like to thank the AE for coordinating the whole review process, and the helpful revision advices for our final submission.